# A Home for the 'Wandering Aramean'—In Germany?

Elke B. Speliopoulos 

Intercultural Studies, Columbia International University, Columbia, SC 29203, USA; elke.speliopoulos@ciu.edu

**Abstract:** Migration to Germany has been a fact of life for the average German since the 1960s. Immigrants started arriving from countries like Turkey, Spain, Greece, or Italy as a post-war labor force was invited to Germany to address workforce shortages. Many of these immigrants ultimately brought their families to live in Germany. One group of these newcomers was Aramean families of Syriac Orthodox faith, forced to flee the Tur Abdin region in southeast Turkey via Syria, Lebanon, and Northern Iraq. This paper will discuss the background and impetus for moving to the West for this immigrant group in detail. It will review the impacts on the life of devout Syriac Orthodox families while living in Germany, a secular country. It will also take an initial look at whether evangelical communities in Germany can come alongside this group, still suffering from a different kind of persecution: the "otherness" of living in Germany.

**Keywords:** Aramean; Sayfo; Aramaic; Germany; immigrant; Syriac Orthodox; secularization; Missiology

## 1. Introduction

Most of us who have studied Scripture probably have caught the 'Wandering Aramean' in Deuteronomy 26:5—hidden in the 613 *mitzvot* (Eisenberg 2004, p. 515) or commandments —when God instructs the Israelites: "You shall answer and say before the Lord your God, 'My father was a wandering Aramean, and he went down to Egypt and sojourned there, few in number; but there he became a great, mighty and populous nation".[1] Who are the Arameans referred to here? And what do they have to do with Germany?

Beginning with a look at the history, background, language, and religion of the Aramean immigrants who began to arrive in Germany in the 1960s, this paper will provide an initial view into the origin and history of the Aramean people of Syriac Orthodox faith living in Germany. Migration patterns and religio-cultural adaptation processes in the context of the Aramean diaspora in Germany will also be reviewed (the study is still ongoing). The assumption at the beginning of the study was that Aramean youths and young adults' attendance is declining in Syriac-Orthodox churches. Patterns become noticeable when reviewing how Aramean parents and grandparents pass their faith to the next generations. The paper will also show how German churches can help them in this effort. This study focused on Evangelical churches due to an assertive youth outreach in most Evangelical Christian communities. The study uses a general hermeneutic approach as the underlying methodology— applied to several historical and secondary sources— and provides a narrative of this ethnoreligious group over time with special attention to the migration process and diaspora situation in Germany. Additional information was gathered during interviews with and surveys of Aramean men, women, and youths in Germany.

As with every research in the academic realm, some key terms require definition. There will be three terms in everyday use for immigrants from the part of the world known as ancient Mesopotamia. Aramean immigrant groups originate from this area, mainly modern-day southeastern Turkey, northern Iraq, and northern Syria. Many people somewhat interchangeably use the terms *Aramean*, *Assyrian*, and *Chaldean* (Aramean Democratic Organization n.d.). Each of these titles describes people that originated in the ancient

Mesopotamian world. Much more will be said later, but briefly, religious immigrants choose to be called *Aramean*, whereas the more politically or future homeland-focused immigrants prefer *Assyrian*. *Chaldean* is mainly used for members of the Chaldean Catholic Church, an Eastern Rite church with immigrants from northern Iraq, northwestern Iran, and southeastern Turkey. It is mentioned here because of the confusion with the plethora of terms describing immigrants from ancient Mesopotamian lands.

*Arameans* discussed here are of the *Syriac Orthodox* faith. The *Syriac Orthodox Church* is named after the Syriac language, the church's liturgical language. This also differentiates it from the term *Syrian*, primarily used today to describe things or concepts aligned with the Syrian Federation.

*Suryoye* or *Suryaye* are the New-Aramaic terms to describe Arameans. These terms are used in the Aramean community today.

*Syriac* is a Semitic language, which originates in the ancient city of Edessa (modern-day Urfa in south-east Turkey) (Butts 2011). It is written in an alphabet derived from the Aramaic language seen in parts of the Old Testament books of Daniel and Ezra. It is the liturgical language of the Syriac Orthodox church.

*Sayfo* (lit. 'sword') is used for the genocide and deportation of Syriac Orthodox Christians from their homelands (Brock 2011). It is also known as the *Assyrian Genocide*. Beginning in 1914 but culminating in a genocide in 1915, Ottoman troops attacked Christian villages and killed their inhabitants. These massacres coincided with the more well-known Armenian genocide and ultimately led to today's diaspora communities.

*National identity*, considered in conjunction with a biblical theology of migration, has been defined as "a means by which culture is defined through these bounded, essentialized notions of 'being' (Tolia-Kelly 2009, p. 260). It is important to note that there is a significant difference in the meaning of the related term *nationality*.

## 2. Immigration to Germany

Watching the news, an observer might conclude that the topic of immigrants is relatively new and limited to our lifetime. Of course, those who know the Scriptures realize that migration is a theme that runs throughout its pages. From Adam and Eve having to leave the Garden after the fall[2] to Priscilla and Aquila fleeing Rome after being forced to leave by a decree of Emperor Claudius,[3] the biblical narrative is replete with migration examples. Studying history, even with minimal effort, one can see that migration has happened throughout human history. People are on the move—at times for economic reasons, but often because they are forced to leave their homeland, frequently due to war or famine.

Newcomers to western Europe come from cultural backgrounds very different from their new neighbors. Many have a divergent religious upbringing than the predominantly Christian European receiving countries (Statista 2022). Even if an immigrant holds to the Christian faith, their faith background and expression may differ from their new home.

Focusing on the Christian new arrivals' example, how does an immigrant who shares the same faith denominator of "Christian"—but not the same denomination—experience life in a new environment? How do they settle in a Western European country, specifically Germany, that has become primarily secularized? How do their descendants live out (or not live out) their faith?[4] How might Germany be different from their country of origin?

Germany is an economically and politically prosperous European country that has risen from literal ashes after World War II to become a European and global powerhouse. It has evolved into an economic model for other nations (White 2010), despite an almost immeasurable impact on its society, infrastructure, and industry from wartime destruction and Nazism, including the genocide of Jews under Hitler.

Germany has had an influx of various nationalities over the decades for several reasons after the initial rebuilding after the war ended. Much of this early recovery was kindled by the Marshall Plan (Hein 2017), which intended to revive and vitalize the industry destroyed in Germany during the war. A significant labor force was needed to put together living

spaces and support the industrial recovery quickly after so much destruction, as many German workers had been displaced. Tamás Vonyó Bocconi explains that

> [r]ecent historiography accepted that industrial equipment and plant had survived the war with remarkably little damage and argued instead that economic recovery was hindered by a combination of factors that disrupted the chains of production . . . [T]he economy remained dislocated after 1950 due to labour misallocation that resulted from the wartime destruction of urban housing (Bocconi 2014, pp. 129–30).

Despite all these hindrances, by the early 1950s, as Kees van Paridon describes, Germany moved into

> a period of unprecedented economic growth starting right at a time when any possibility for economic growth seems utterly impossible. At the end of World War II, Germany was regarded to be in that situation. The country was greatly weakened by an enormous physical devastation, loss of life, the division of the country into two parts, and the allied occupation. That such a country could experience a dramatic postwar economic revival seemed out of the question. It seems therefore quite understandable that the very strong economic recovery of the Federal Republic of Germany in the fifties and the sixties was seen as an economic miracle. (van Paridon 1998, p. 651)

To augment the German labor force, Turkish, Greek, Spanish, Yugoslavian, and Italian guest workers came to Germany in the 1960s to support this booming economic development. Many of the workers eventually brought their families and stayed in Germany. A majority of migrants to Germany came because of the moneymaking opportunities offered. Some, however, had lived through persecution in their countries of origin and sought safety and the chance to provide for their dependents. These immigrants brought with them various faith traditions, e.g., Islam. To cite one example, Afghan refugees came in several waves, starting in 1979 with the Soviet invasion of Afghanistan, and continuing in an ebb and flow based on political events in the country (Fischer 2019). After the fall of the Berlin Wall, other nationalities joined the German potpourri of humankind: Russian, Polish, Germans from former East Germany, and Albanians, among others. Over the past decade, one million Syrian refugees have enriched the German cultural landscape. Now with the Taliban ruling Afghanistan, there is a significant rise in the number again of Afghan refugees. Many Germans struggle with the ever-expanding diverse landscape.

### 3. Who Are the Arameans?

Arameans are by religious and cultural background Christians of Syriac Orthodox faith. They originate from primarily south-eastern Turkey, specifically a region called Tur Abdin (or "Mountain of the Servants [of God]"). Adam Becker describes the area from where the families came as

> " . . . an elevated plateau with a microclimate reminiscent of the Mediterranean, although it is hundreds of kilometers from the sea. Today the Tur 'Abdin, part of contemporary Mardin province, is considered the spiritual homeland of the Syrian Orthodox (West Syrians, Süryani), the East Syrians' centuries-long rivals, most of whom now live in Istanbul or scattered across northern Europe and elsewhere. From the enchanting heights of Mardin, the city on the southwest corner of the Tur 'Abdin, one looks south into the flat plains of Syria". (Becker 2015, p. 44)

Regarding ethnicity, many/most Arameans do not perceive themselves as Turkish but as a distinct ethnic group. Instead, they see themselves as the descendants of those who inhabited that portion of ancient Mesopotamia. They speak a Central Neo-Aramaic dialect of Aramaic, Surayt, also called Turoyo (Talay 2002). On the other hand, their church's liturgical language is classical Syriac, another Aramaic dialect (Syriac Orthodox Resources 2001). The Aramaic language will be given a closer look later.

In the 1960s, the first Aramaean immigrants—primarily men—arrived in Germany. The legislation in Germany had shifted to allow so-called "guest workers" to come to Germany with full work permits (Oltmer et al. 2012, p. 10). After immigration initially was limited mainly to *Aussiedler,* or ethnic Germans that were often forced to flee eastern lands and internal refugees fleeing the socialist German Democratic Republic, this changed, as Eule writes, "from the mid 1950s, when imminent labour shortages spurred the West German government to recruit temporary 'guest' workers. As a result, bilateral agreements were made with Italy (1955), Spain (1960), Greece (1960), Turkey (1961), Portugal (1964), and Yugoslavia (1968) through which some 2.6 million mainly male low- and un-skilled workers arrived in the FRG"[5] (Eule 2014, p. 10). And so, in 1961, Turkey joined the other nations that were allowing workers to apply for positions and travel to Germany, filling jobs that had gone unfilled, many in, e.g., agriculture or mining.

Unlike some other—but not all—immigrants during that time who may have primarily arrived due to excellent economic prospects compared to their countries of origin, Sofia Mutlu-Numansen and Ringo Ossewaarde write about the immigrants from southeast Turkey:

> Without a state or intellectual elite of their own, and being deprived of their means of expression in post-Ottoman Turkey, these forgotten peoples, had no means for recognition. After another wave of persecution in the 1970s, many of them managed to obtain political asylum in Western Europe, particularly in Sweden, Germany and the Netherlands. (Mutlu-Numansen and Ossewaarde 2015, p. 430)

Arameans arrived in Germany due to the after-effects of a more somber reason: a genocide their ancestors had survived and continued persecution in their ancient Mesopotamian homeland.

## 4. The Sayfo

Thanks in part to the success of the 2017 Hollywood movie "The Promise," starring Christian Bale (George 2016), more people, in particular in the United States and Europe, have heard of the Armenian genocide—the systematic destruction of Armenian people by the Ottoman Empire during World War I. Many descendants of Armenian survivors live in the United States of America as well as other diaspora locations. One cannot speak of the Aramean people at discussion here (and be careful with the similarity in the sound of the name when quickly listening) without considering what drove them to leave their homeland—much like the Armenian people. As Mutlu-Numansen and Ossewaarde note,

> [g]enerally, there have been few publications on the Syriac Sayfo, as contrasted with research on the Armenian genocide. With some notable exceptions (Gaunt 2006; Khosoreva 2007; Omtzigt et al. 2012; Travis 2011), the Sayfo genocide is typically mentioned within the context of the Armenian genocide, lumped together with the mass killings of Armenians, Yezidis and Kurds in what Levene (1998) calls the 'zone of genocide'. (Mutlu-Numansen and Ossewaarde 2015, p. 430)

To give the historical review of the genocide some depth, in two phases, once in 1895 and then again in 1914/15, the political rulers of the countries intended to gather Muslim peoples and strengthen their national identity. In 1912, the Christian peoples gained their independence, but the turmoil of the First World War and the losses meant that the Ottoman Turks resolved to expel the Christian peoples to strengthen their territory. How do you get rid of non-Muslims? In this case, a Holy War—*Jihad*—was declared in 1914, claiming the Christian peoples to be enemies and traitors.

Mustafa Afsakal writes about this time,

> By the nineteenth century, however, the challenges presented by Ottoman subjects linked to foreign powers became all-pervasive. As a result, religion became intensely politicized at the state level. Muslim-Christian relations lay at the very

heart of Ottoman state policy. How could the state maintain the empire's regions with large Christian populations in the Balkans and parts of Anatolia under the Ottoman umbrella? . . . The Young Turks who came to power in 1908 and more firmly in 1913 also understood the power of this ideology, and they went on to capitalize on it in 1914, when they entered World War I on the side of Germany and Austria-Hungary and—despite the apparent contradictions that alliance with the Christian powers entailed—declared jihad on 14 November 1914. By this time, they had embraced the belief that the empire could not be salvaged by reforms, diplomacy, or "those old books of international law, but only by war". (Afsakal 2012, pp. 287–88)

Initially, a plan was worked out to get rid of Christians: they should pay gold to avoid being drafted into the war. All Christian males, Arameans, Armenians, Catholics, Chaldeans, and Protestants aged 20–45 should pay fifty dinars of gold. Such a coin was minted from 4.25 grams of 22-carat gold. To give a comparison, today's value (in 2022) would be roughly $110,000—in other words, a price impossible to pay for the villagers.

Shortly after that, however, the actual plan came to light. Beginning in 1914 but culminating in a genocide in 1915, Ottoman troops attacked Christian villages and killed their inhabitants. Ottoman forces persecuted those they deemed 'other.' This meant that discrimination was also exerted against the Christian citizens of the Tur Abdin region, among them the Syriac Orthodox Arameans. In particular, the bishops, priests, and monks, but also people who had been given a good education, were rounded up, tortured, and then killed. One of the methods used was starvation. The Ottomans demanded from the villagers that wheat, sheep, coal, and other items be given to feed their troops. The situation continued to escalate, despite all efforts to return to peace. These massacres are still only partially recognized and were a driver of immigration to Western Europe, leading to today's diaspora communities. Between 1.6 and 2.7 million Christian Armenians, Arameans/Assyrians/Chaldeans, and Pontic Greeks died during this terrible time in the region.[6]

Boutros Touma Issa, in a collection of readings collected from family members who originated in Mesopotamia and are members of the Syriac Orthodox Church of Antioch, leaves no doubt about the horrors of the genocide:

> The atrocities of those deceivers and fraudsters went further where several of the bishops, priests, community leaders, and other children of the community were either suffocated with smoke, or burnt alive, or cut in pieces, or buried in sand while still alive, or dropped in moving sand, rivers, or lakes to face their death without any mercy. In addition, there were those who joined the martyrs either dying from hunger or cold, with their bodies left to feed the birds and the beasts. (Touma Issa 2017, p. 126)

In some districts, Mutlu-Numansen and Ossewaarde write, "more than 90 percent of the Christian population was killed" (Mutlu-Numansen and Ossewaarde 2019, p. 413). More than 500,000 people of Aramean, Assyrian, and Chaldean backgrounds were killed in southeast Turkey. This genocide of Syriacs is often lumped together with the Armenian genocide, yet it deserves its own recognition. The term *Sayfo* (lit. 'sword') continues to be actively used within the Aramean community for the genocide and deportation of their Syriac Orthodox Christian ancestors and other Christian groups from their homelands (Brock 2011).

The almost complete decimation of the Aramaic educated and clerical elites left behind a small number of families in Tur Abdin. They did not have the educational background or financial means to significantly impact their lives in this region of southeast Turkey. Anecdotally and to provide real-life stories as collected in interviews, as one man shared, his small village did not even have a school until 1965.[7] Another man in the Syriac Orthodox community in Germany shared that he attended school for grades 1 through 5, plus parochial school to learn the Syriac language, prayers, melodies, and history. Only families who had sufficient money could send their children to higher grades. He left the Tur Abdin

region when he was 14, traveling to his new home via Istanbul to join his two older siblings who were staying with relatives already in Germany. His family arrived in Germany in 1980 in a staggered manner. His parents sent the older children first, having learned of more significant economic opportunity and a lack of persecution there. They then followed with the two youngest children. It was becoming increasingly complex for Christian families to live in Turkey due to continued attacks by Kurdish and Turkish assailants and the lack of opportunity to earn a living. There was "no future, no higher education, no universities, no employment".[8] Upon arrival in Germany, his education did not continue except for two German courses. Through sheer diligence, he was able to self-educate and is now the respected and sought-after author of currently 22 published books being used in the global Syriac Orthodox community. He is a deacon in his church and a Syriac Orthodox religion teacher in the German school system.[9]

The collective memory of *Sayfo* has been carried forward through the generations and is an important cultural aspect of the *Suryoye* self-understanding. Every 15 June, *Sayfo* commemorations are held throughout Aramean communities in Germany. Memorials proposed or built in cities around Germany still receive Turkish government pushback today[10]. The trauma experienced is something that is carried forward in each following generation. Ciano Aydin writes about collective trauma:

> Several studies confirm long-term effects of trauma caused by genocide. Descendants of the 1994 Tutsi genocide survivors in Rwanda were at high risk of developing mental health problems, had a high trauma load, and missed family integration and support even 16 years after the genocide (Rieder and Elbert 2013). In other studies, descendants of Armenian and Syriac Christian genocide survivors expressed feeling burdened by having to carry emotional memories of previous generations. They indicated that they have more difficulties living a normal emotional life, which is expressed in deep sadness, distrust of outsiders, and a damaged sense of their identity and reality (Cetrez 2017; Kalayjian et al. 1996). Recently, Yehuda et al. (2015) indicated that children of genocide survivors can even inherit trauma in their DNA. This study revealed that Holocaust offspring had 7.7% lower methylation than control offspring, and had low cortisol levels associated with depression, emotional hypersensitivity, and social anxiety. If a traumatic event shared by an ethnic or religious group is not dealt with or paid proper attention, it not only can cause shifts, disruptions, and disturbances in the group's cultural identity and hinder its ability to flourish in the future, but it ultimately can also lead to its obliteration. (Kalayjian and Weisberg 2002; Aydin 2017, p. 128)

## 5. How Is the Aramean Faith Different?

The Syriac Orthodox faith of Aramean families is unfamiliar to most Western Christians. Often lumped together under the "Orthodox" title, upon inspection, one finds that there are fundamental differences. For most people of the Christian faith in the West, "Orthodox" comes mainly from Eastern Orthodox expressions: primarily Greek Orthodox or Russian Orthodox. The Syriac Orthodox Church points its identity back to Acts 11:26 when Saul (Paul) and Barnabas met in Antioch with the believers and taught the church. There the believers in Antioch were first called Christians. As Heidi Armbruster notes, "[s]cholars generally locate the historical origins of Syriac Christianity in both a Jewish Christian heritage and the gentile Christian milieu of Antioch, one of the major centres of early Christianity" (Armbruster 2013, p. 6).

Any student of church history will point to the year A.D. 1054 when the East and the West churches split, better known as the Great Schism (Galli 1997). A lesser-known fact is that an earlier schism occurred in the 5th century that split the Orthodox world into the Eastern Orthodox churches and the Oriental Orthodox churches (Penn et al. 2022, pp. 11–13). One of these Oriental Orthodox churches is the Syriac Orthodox Church of Antioch. The others are the Coptic Orthodox Church of Alexandria, the Armenian Apostolic Church,

the Malankara Orthodox Syrian Church, the Ethiopian Orthodox Tewahedo Church, and the Eritrean Orthodox Tewahedo Church. The starting point of the ultimate schism was the differing Christological understanding of two men: Cyril of Alexandria and Nestorius of Antioch, explicitly concerning their respective interpretations of the two natures of Christ. Already in AD 431 at the Council of Ephesus, Nestorius's position to refuse to call Mary *Theotokos,* the God-bearer, but rather only allow her to be called *Christotokos*, the Christ-bearer, brought the Antiochian position to an extreme. This position emphasized the "distinctiveness and integrity of humanity and divinity in Christ to such a degree that they could not easily affirm a true unity of the two in the single person of Christ (FitzGerald and Gratsias 2007, p. 11). Language issues also burdened the discussion between Cyril and Nestorius. Nestorius was ultimately declared a heretic, which several modern historians and theologians now see as primarily a political decision.[11]

The Oriental churches supported Cyril's position and felt that the Chalcedonian definition was too Nestorian. As Father V.C. Samuel writes,

> At the same time, the Alexandrine tradition and particularly Cyril had a great hold in the east, and the synodal committee which drew up the definition had men who would stand by it. These men succeeded in putting in a few emphases coming from their tradition in the council's definition, which enabled sixth century Chalcedonian theologians in the east to develop a doctrinal position which was as anti-Nestorian as, if not more anti-Nestorian than, that of the council's opponents. (Samuel 2001, p. 110)

The authors of the excellent *Syriac Orthodox Resources* website, among them the well-known researcher and publisher George Kiraz, further clarify,

> The Council of Chalcedon in A.D. 451 resulted in the schism of Christendom into two groups. The Catholic (Rome) and Greek (Byzantine) Churches accepted the Council, while the Syrian (Antioch) and Coptic (Alexandria) Churches rejected it. The former group adopted the doctrine that Christ is *in* two natures, human and divine, while the latter adopted the doctrine that Christ has one incarnate nature *from* two natures. It is worth noting that the drafts of the Council were according to the position of the Syrian and Coptic Churches. The final resolution, however, was according to the doctrine of the Western Churches. The difference lies in one preposition as explained. (Syriac Orthodox Resources 2004)

Meanwhile, another political conflict was brewing between Constantinople and the Oriental churches. Deacon Hanna Aydin of the Syriac Orthodox Church writes:

> The Council of Chalcedon was dealing with a new paragraph (can. 28) in which the power of the Patriarch of Constantinople over the Orient was to be expressed. Since the Byzantine Emperor exercised power over the Orient, the Patriarch of Constantinople was also to exercise sole power over the Oriental Churches. When the Oriental Churches (Syrian, Egyptian, Armenian and Ethiopian) legitimized themselves as older and rejected the Constantinopolitan jurisdiction, they were accused with the word 'Monophysitism' in order to be able to persecute them; in reality, it was all about political interests. (Aydin 1990, p. 45)[12]

While Christological considerations were still prominent, this point weighed heavily on the decision of the Oriental Orthodox churches to ultimately not affirm the declarations at the Council of Chalcedon in AD 451. Despite efforts by Emperor Zeno in AD 482, who issued a conciliatory document known as the *Henotikon*, which affirmed the creed of Nicea, the rift was irreparable. Opposing voices arose primarily out of Egypt and ultimately caused this effort to fail (Samuel 2001, pp. 143–50). This triggered the final schism of the miaphysite—a term suggested by Dr. Sebastian Brock of Oxford University to accurately describe the Syriac Orthodox position (Syriac Orthodox Resources 2000)—or non-Chalcedonian churches, as they are sometimes referred to, from the dyophysite churches or Chalcedon-affirming churches.[13]

Moving forward in time, Herman Teule studied the *Suryoye* concept of self-identity in the Middle Ages by examining three authors—Dionysius bar Salibi, Jacob bar Shakko, and Gregory Barhebraeus—who wrote during the 12th and 13th centuries, a period known in this part of the world as the Syriac Renaissance (Teule 2009, p. 179). In particular, Teule wants to demonstrate the impact of Arabic on the Syriac community and the attempt to keep Syriac alive. He does so by studying the writings of these different but contemporary authors. Teule puts this in the context of identity formation within the Syriac community. As he highlights, all three writers are "considered by present-day Suryoye as bearers of their identity" (Teule 2009, p. 180).

Teule looks deeper into three areas of the writings of bar Salibi, bar Shakko, and Barhebraeus: their engagement with the Islamic cultural world, the religion of Islam, and the topic of divisions between Christian groups. Teule highlights that the three men were still proposing a distinct Christology vis à vis their neighboring Christian denominations. Bar Salibi refuted a West Syrian monk named Rabban Yeshu'. This monk had reflected on these inter-Christian struggles: "Is it right to consider only ourselves as orthodox and the others as heretics? Is it right to constitute ourselves the judge of other Christians" (Teule 2009, p. 187)? Bar Salibi contended that Chalcedonians and Nestorians were to be refuted because their Christology differed, and their liturgical practices set them apart. Bar Salibi even described his vehement opposition to Armenian Christians, although also miaphysite, as their practices again were different. He strongly wanted to protect his community. Both bar Shakko and Barhebraeus were likewise interested in preserving the Christological formula of the Syriac Christian community and demanded that the search for truth was only to be completed in their community (Teule 2009, pp. 186–87).

Today, the miaphysite churches are considered the Oriental Orthodox Church—non-Chalcedonian, whereas the others are dyophysite in nature—or Chalcedon-affirming.. They are the Eastern Orthodox churches and the Western churches. The Syriac Orthodox church is also called the 'Syrian Orthodox' church; however, recent political developments in Syria have made the use of 'Syriac Orthodox' clearer (Syriac Orthodox Resources 2004). Sometimes, the term 'Jacobite Church' is also used. However, this is considered derogatory by the Syriac Orthodox faithful, who point to Christ as the church's founder. This misnomer is based on the work of a dedicated monk, Jacob Baradaeus. In the 6th century, through the support of Empress Theodora, he became the general metropolitan, rebuilt the decimated Syriac Orthodox clergy, and was tasked with restoring the persecuted churches of Antioch and Alexandria (Standing Conference of Oriental Orthodox Churches 2022).

Bringing the two Orthodox worlds back together seems to be a Herculean task. Theologians and church leaders have made many ecumenical efforts to reunite the Eastern Orthodox and Oriental Orthodox churches. Yet, a significant amount of pushback has come to agreements made in the past, particularly from the Holy Monastery of Saint Gregory on Mount Athos (Orthodox Christian Information Center n.d.).

## 6. The Aramaic Language and Its Dialects

A walk down the Semitic language tree of Aramaic, an ancient language, shows it has numerous dialects and has been in continuous use since the 11th century BC. George Kiraz writes that it "became the lingua franca of the Near East by the 6th century BC. It was the native tongue of the ancient Chaldeans, a second language to the Assyro-Babylonians, an official language of the Persian Achaemensians, and a common language of the Jews, replacing Hebrew. Jesus and the Apostles spoke and preached in Aramaic" (Kiraz 2013, p. xxi). Its spread was mainly due to Christianity's spread in the Semitic-speaking world and the Silk Road (Kiraz 2013, p. xxi).

Syriac is another dialect of Aramaic that began in the Edessa area, present-day Urfa in Turkey. It is prominent as the liturgical language of the Syriac Orthodox church. Many ancient documents are written in Syriac, including theological writings. While Syriac declined in prominence between the 14th and 19th centuries, Syriac has retained importance as the liturgical language of Syriac Orthodoxy (Kiraz 2013, p. xxi). Because of this history,

the Arameans, by their language and many other achievements as theologians, thinkers, and poets, have a long and proud part in the history of humankind.

Jehu Hanciles' 2021 book *Migration and the Making of Global Christianity* describes the shaping of Christianity from its early days through the end of the medieval period. He discusses the contentious space between the Roman Empire and the Persian Empire and the Christians caught in the middle. Hanciles describes the critical importance of the Syriac language and confirms that it served as a "lingua franca, only being replaced by Arabic after 637" (Hanciles 2021, p. 216). Hanciles showcases the importance of the language, not just in ecclesiastical terms for the Syriac Orthodox Church, but also as the language that served the entirety of Syrian and Mesopotamian lands, thereby making it "a lubricating ingredient that facilitated trade and made communication and countless transactions between vastly diverse populations possible" (Hanciles 2021, p. 216).

In everyday life, the earlier generations of Aramean immigrants speak Surayt, or Turoyo, another dialect of Central Neo-Aramaic (Oez 2018, p. 340). This language was orally handed down and therefore lacked a written system. It originates in Tur Abdin and is considered severely endangered on UNESCO's *Atlas of the World's Languages in Danger* (Moseley and Nicolas 2010). Most Turoyo speakers have immigrated from the Tur Abdin region to Western Europe. While about 20,000 people lived in the Tur Abdin in the 1960s, today, there are only about 2000 (Talay 2002, p. 69). As Shabo Talay notes, the children of these first-generation Arameans learn the local language (in this case, German) as soon as they enter Kindergarten at the age of three or four. This means they begin mixing languages in their speech, known as *code-switching* (Talay 2002, p. 70). The churches presented elements, e.g., Scripture readings, during the Holy Liturgy to the people in Turoyo, which led to the conviction that teaching Turoyo was the church's role. However, the *madrashyote* (also called *madrasse*), or church schools, only teach Syriac for liturgical purposes. Some efforts have taken place to add a written system for Turoyo. Since many children and youths do not know the Syriac letters, a concept was discussed to use Latin letters potentially. However, this would lead to a further decline in the use of Syriac as the liturgical, written language (Talay 2002, p. 75). Thus the Turoyo language continues to decline with the younger generations (Talay 2002, pp. 71–72).

An anonymous, written survey of 34 school children 13–15 years old provided by a proxy—in this case, a religion teacher in local schools—showed that almost all respondents stated that their parents had taught them Turoyo, and they spoke it themselves. Yet, the mention of a *Madrasse* (school) for children who need to learn outside the home shows that this sample may not be representative. More research is required to assess this independently outside a religious instruction setting.[14]

## 7. Aramean or Assyrian?

One might hear of this people group referred to as Assyrians as well. Both terms are used to describe the people group that calls themselves *Suryoye.* Both may very well originate from the same area and speak the same language, yet each group has differing motivations and goals. An ancient people is being used to highlight the quest for national identity, as Adam Becker asserts:

> The name Assyrian as used for the contemporary ethnoreligious community of Assyrians is an "invented tradition," a retrieval of an ancient appellation that had fallen into disuse for over two thousand years. Invented traditions "are responses to novel situations which take the form of reference to old situations, or which establish their own past by quasi-obligatory repetition". The use of Assyrian derives from Western sources, not from a continuity of identity between the ancient Assyrians and the modern ones. (Becker 2015, p. 299)

It was only in the 1970s—already in Western Europe—that differentiation was desired. Mutlu-Numansen confirms the application of a different term:

Though the Arameans and Assyrians are technically two different peoples, their Sayfo victims in Turkey regarded themselves, and were regarded by others, as one people: Syriacs (from the Syriac-Orthodox religion so intertwined with their identity). It was only after migration from Turkey in the 1970s that these Syriacs started to identify as *either* Arameans or Assyrians. (Even within one family, people can disagree on their identity). (Mutlu-Numansen and Ossewaarde 2019, p. 413)

In *Soccer & Society* journal, Carl Rommel writes on performative spaces in the football (soccer, here) world of *Suryoye* migrants in the Swedish town of Södertälje with a large *Suryoye* population (Rommel 2011). In *Playing with Difference: Football as a Performative Space for Division Among Suryoye Migrants in Sweden*, Rommel interviewed male soccer fans who are fans of one of two *Suryoye* soccer clubs. The two clubs demonstrate the internal conflict within this immigrant group from the southeastern part of Turkey. In Tur Abdin, these families were solely defined by their Christian religion. They were suddenly called 'Assyrians' (Assyrier in Sweden) in Sweden.

This difference in understanding resulted in two very prominent soccer clubs, one ascribing the Assyrian and the other the Syriac descriptor to their name and identity. Rommel reviews this experience in light of identity formation. He shows that while Assyrians assign a political aspect of originating from the Assyrian Empire of biblical days to their identity, the Syriacs focus on the Aramaic language and the history of the Syriac Orthodox church. While they both use *Suryoye* to describe themselves, these deeper-reaching identity aspects write a story where "history ... divides rather than unites" (Rommel 2011, p. 854).

*Suryoye* identity envelops both the Assyrian and the Aramean characters. Regardless, there are still disagreements as to what is relevant. As an example, from a conversation with the leader of an Aramean group in Germany, roughly one-third of the Syriac Orthodox church members supported the Assyrian New Year on April 1st enthusiastically and saw it as an identity-relevant celebration, while about two-thirds still thought that this celebration was not a part of the Aramean culture.[15] In this paper, Aramean is the preferred term as the Syriac Orthodox men, women, and youth interviewed chose this as their self-descriptor.

## 8. Aramean Family Structures

While there is a small amount of intermarriage in the Aramean community, most marriages are conducted within it. Interviews with Aramean men and women in the context of Syriac Orthodox Church communities showed a powerful religious influence within the family, starting in early childhood when the babies are baptized into the faith. Aramean girls are raised with rigorous moral understandings. While some Aramean men and women are no longer part of the Syriac Orthodox Church, interviews suggested continued adherence to traditional values in raising children.[16]

Divorce is rare since marriage is one of the seven sacraments of the Syriac Orthodox Church, but it is acknowledged in the writings of the Syriac Orthodox dioceses. An example from the Archdiocese of the Eastern United States states, "In our present and modem time, divorce is rampant and on the rise. It is creating havoc and tremendous problems for Christian couples and children. The family values and ties are breaking loose and they are impacting the Christian society in very negative ways" (Ghattas 2010).

Church attendance is also considered a normal part of Aramean life in Germany. The social life in the church communities is very active, as shown through mentioning various events throughout the year by teenagers surveyed. Events such as weddings, baptisms, patronage feasts, and Mother's and Father's Day celebrations allow the community to bond.[17] Equally vital are the mourning expressions in the form of defined mourning period activities in the church for members who have passed away. The church community is always invited.[18]

Several schoolchildren surveyed were also active within the church through altar service (for the boys) and being part of the choir (for the girls).[19] These roles are very

structured within the liturgical setting of the Syriac Orthodox Church (Pircek, Fehime, and Elisabeth Karabas n.d.).

## 9. The Impact of Living in a Secular Society Such as Germany

Often in decades past, Aramean neighbors of Germans were considered just a form of Turkish immigrants. This image is slowly changing, which can be seen in a greater awareness of the Syriac Orthodox Church in public spaces/public life. Yet, some considerations remain pertinent to what living in Germany means to Syriac Orthodox men and women, e.g., the invisibility of this community. Heidi Armbruster describes this vividly in a chapter entitled "Trajectories of invisibility" from her book *Keeping the Faith: Syriac Christian Diasporas.* She highlights the historical plight of the Syriac Orthodox, which led to their present experience in Germany, where they are "largely disappearing in the community of so-called 'Turks'" (Armbruster 2013, p. 118). Armbruster discusses the legal status changes of immigrants to Germany in 2000 that eliminated the longstanding *jus sanguinis* ("right of blood" or birthright citizenship) requirements for German citizenship, making naturalization—at least legally—more accessible (Armbruster 2013, p. 120). Before this legal change, an immigrant's child was not automatically a German citizen, as they would have been in the US based on *jus soli* (right of birthplace).

Her chapter on "Change and generation in Berlin" (Armbruster 2013, pp. 161–82) is particularly interesting. It addresses the second-or third generations and their experience in their now no longer deemed 'new' environment, as it was for their parents and grandparents, but rather—what is now—their native German home. This is where the research interest focuses: what does it mean to raise children in a secular Western European culture when your culture of origin is profoundly religious? The issue is real, as the following statement from the Orthodox Bishops' Conference in 2012 shows: "We are concerned to see how many younger members of the church grow up without the necessary religious instruction to help them maintain the faith of their fathers and mothers and later pass it on to their children"[20] (Gemeinsame Kommission der Deutschen Bischofskonferenz und der Orthodoxen Bischofskonferenz in Deuschland 2012). Understanding how Syriac Orthodox churches are established in Germany is imperative for their interaction with other denominational groups.

## 10. The Syriac Orthodox Church in Germany

Reinhard Thöle writes in the article "Orthodox Churches in Germany: From Migrant Groups to Permanent Homeland" about the interplay between various arms of Orthodoxy in Germany and how Orthodox churches interact with German Catholic or Protestant churches. Understanding the Orthodox landscape in Germany is fundamental, particularly the Christological differences between the Eastern and Oriental Orthodox churches. Thöle states, "Newcomers are looking for churches not merely because they are Orthodox, but sometimes even more with the expectation of finding a national home in a foreign environment" (Thöle 2014, p. 93). Immigrants to Germany started all Orthodox parishes in Germany. They would invite priests from their home nations to join them in Germany and serve these parishes (Thöle 2014, p. 91). This is true across all Orthodox denominations. In Germany, the Syriac Orthodox Church of Antioch has about 100,000 members across more than 60 churches of their own (Erzdiözese der Syrisch-Orthodoxen Kirche in Deutschland n.d.). In the city of Gießen, 45 min north of Frankfurt, and its surroundings alone, there are four Syriac Orthodox churches, three of them in a smaller suburb community of just over 18,000 people. There are a few cities and towns in Germany where larger groups of Aramean families have settled. The Syriac Orthodox Church supplies them well with church buildings and clergy. All of this is funded through the tithes and offerings of the congregants. Unlike the German Catholic and Protestant state churches, Orthodox churches in Germany are self-funded (Praetor Verlagsgesellschaft mbH n.d.).

### 11. National Identity in Light of Scripture and in Historical and Modern Thought

The conversations with interviewees during this study to understand their own self-understanding and/or description almost exclusively ended up in a statement of "I am Aramean". National identity may be difficult to define for a people without a country. Many researchers have attempted to define what "national identity" means, i.e., what makes up a person's self-perspective of whom they are against the backdrop of national distinctiveness and character as applied to themselves.

In his 2020 dissertation, *A Biblical-Theological Study of Geography for Developing Missions Strategy to the Nations*, Matthew Hirt delivers an exhaustive review of what it means to develop a missions strategy for the nations. Hirt begins by reviewing the biblical theology behind the concept of national identity related to land. Across several Old and New Testament passages, he proposes that the authors of Scripture deploy a metonymy in using either the land or the nation to represent the nation's people (Hirt 2020, p. 36). Hirt shows that the loss of land through war meant the loss of national identity. He concludes that "[t]he geographical aspect is the most obvious component of national identity" (Hirt 2020, p. 78).

Andrew H. Kim, in his 2020 book *The Multinational Kingdom of God in Isaiah,* lets the reader look at the anthropological definition of a nation by investigating this concept in the reading of Genesis 10–12. Chapter 4 focuses on Old Testament passages that discuss the idea of "nation" from several angles, e.g., eschatological and consummate. An important takeaway is what Kim writes on what makes a nation: "[T]here is a characteristic that all anthropologists agree that a nation must possess, namely a historic homeland" (Kim 2020, p. 3). When considering immigrants such as the Arameans in Germany, who now have second and sometimes third generations born in Germany, it is critical to view how such a displaced person group can be a nation without such a homeland or how they can function better within their current context of being immigrants to another country.

Dorothea Weltecke's contribution to *Church History and Religious Culture* is an article that focuses on Michael the Syrian, the 12th-century patriarch of the Syriac Orthodox Church, and how his writing—notably his *Chronicles*—and his influence shaped Syriac Orthodox identity. This identity had been under polemical attack during his time as patriarch. Michael advocated primarily for the Syriac-speaking regions of the Syriac Orthodox world. Weltecke highlights how this Syriac Orthodox identity was crystallized by the term mhaymne, the believers. The term points to the Syriac Orthodox community's self-understanding that a believer is very worthy-of-trust and honorable. It also focuses on the person's religious identity addressed by this term: "Someone who apostatized to Islam or to Greek Orthodoxy stopped being *mhaymno.* At the same time, he also stopped being *Suryoyo*, as Michael did not see them as a member of his group any more" (Weltecke 2009, pp. 117–18).

Orthodoxy and what it meant to be *Suryoyo* were topics where Michael sought understanding, gave defense, and pleaded his ethnic case. Michael argues that the original language spoken by humankind in general before the Tower of Babel was Aramaic, in one example of his writing. Michael wanted to identify with the ancient Near East. It was essential since this identity was not threatened so much by Islam but "by the quarrel between the churches, more precisely, the quarrel with the Greek Orthodox and their attack on Syriac Orthodox identity" (Weltecke 2009, p. 120). Attacks on Syriac Orthodox identity often combined and collapsed religious and historical episodes. Weltecke writes, "The most important factors are similar to the present-day situation—polemical questions and attacks from outside" (Weltecke 2009, p. 122), something she elsewhere puts into a modern-day context: "One important challenge is the under-representation of Syrians in the narratives of the history taught at school and covered in the media in Germany" (Weltecke 2009, p. 116).

While Weltecke's article focuses on Michael the Syrian, a historical person from the 12th century, much of what she writes applies to the discussions in the Syriac Orthodox community today. Whether it is belittling remarks from those from Turkey but outside

the Suryoye community or other exclusion from German society, Syriac Orthodox migrant families experience much the same in the early 21st century in Germany.

A contemporary look at the meeting of religion and an understanding of national self-understanding comes in the 2022 article by Stephanie N. Shady, "Territory and the divine: the intersection of religion and national identity". Shady takes a look at the impact of religious behavior on national identity. She compares devout faith, expressed through religious adherence, to a more Christmas/Easter type of faith expression and, finally, to the complete repudiation of any religious affiliation in Europe. Shady discusses various studies linking political leanings to religious expression, producing ethnocentrism and prejudice (Shady 2022, p. 746). She shows that social standing and voting behavior are closely tied to current or even former religious belonging despite secularization. Setting this in context to immigrant religions, Shady observes that religious affiliation of any level from the majority group drove a higher national identity. This can result in religious minority groups becoming "antithetical to membership in the country" (Shady 2022, p. 750). Shady briefly discusses secularization in children but does not tie this to a particular group, i.e., German-born or immigrant. In concluding her research, Shady suggests that "secularisation does not erase the influence of religion in politics" (Shady 2022, p. 761).

For the Aramean people in Germany, language also appears to be a key element of what makes up their perceived identity. This seems to run counter to today's culture of leveraging certain key languages, e.g., English, as a modern *lingua franca*. Anthony D. Smith, in his classic *National Identity*, discusses the role of a *lingua franca* in days gone by vs. today:

> In the high middle ages, Latin and Arabic achieved a genuinely trans-territorial and trans-cultural sway. But, in those cases, there was a corporate identity—the medieval clergy and ulema—with a transterritorial function that a lingua franca could serve. . . . Today, with many 'low' cultures turned into literal 'high' cultures for mass, standardized public education, national languages have replaced the earlier lingua franca. But not entirely—the extension of certain prestige languages to facilitate communication and exchange over wide areas has promoted a sense of loose cultural kinship within culture areas and sometimes even beyond them. (Smith 1991, pp. 172–3)

The Syriac liturgical language and the Turoyo colloquial spoken in the homes was never used for public education, rather only in ecclesiastical settings, in their Turkish villages. It served as a vehicle to set themselves apart in a hostile society and preserve their ethnic and national identity understanding. Even today in the diaspora, it serves as a buffer to this modern concept of leveraging the perceived *lingua franca* to become part of the broader cultural context of Europe. However, as language skills slip in the younger generations, this standalone aspect will impact the understanding of national identity in the younger generations without doubt. This will also cause disruption between first- and often even second-generation Arameans and their offspring in the Aramean community as the decline of the Syriac and Turoyo languages impacts both the Syriac Orthodox Church and the remaining hope for a future Aramean homeland.

## 12. Evangelical Communities in Germany in Relation to the Aramean Community

The question of how evangelical communities in Germany might come alongside Arameans, especially their children, teenagers, and young adults, motivates much of this research. Having seen how little impact faith has overall on German society at large in the 21st century, it is an imperative missiological consideration to find ways to re-introduce a younger generation to the Christian faith. Many Evangelical German churches seek to do this within their target groups, either their offspring or immigrants who have come to Germany.

The Free Evangelical Community (Freie Evangelische Gemeinde, or FEG) church in Gießen offers simultaneous translations into six languages: English, French, Russian, Turkish, Farsi, and Arabic, and an English-speaking small group on Sunday nights.[21]

However, these are targeted at immigrants from other people groups without a Christian background, such as those from Iraq or Afghanistan. An interview with the pastor showed that the church is very active in missions and outreach, yet it does not engage with the Syriac Orthodox community. It has refugee programs, which seek to address more recent migrants, most still in the Gießen refugee center. In its youth programs, the pastor had an excellent insight: their youth programs worked when youths from outside the church community attended together with the children of the families in the church.[22] This may be a helpful consideration in the Aramaean community. Their juveniles are straddling the two worlds of church and school/friends outside the church. An approach that brings youth from outside the church to youth activities may generate interest but may also spark fear in the parents as they might see their children in danger of meeting and ultimately marrying someone outside the community. This pastor also desired to understand the self-perceived role of the Syriac Orthodox Church in Germany.

One other interesting aspect raised by the Evangelical pastor was the diversity he seeks for his church community. When experimenting with English-language worship songs to begin to demonstrate the cultural heterogeneity already existing in the church, the leadership team quickly realized that neither the targeted immigrants nor their older German worshippers could understand the lyrics of the songs. Instead, he is setting out to attempt a new experiment: having a church council that is entirely made up of a mix of Germans and the various immigrant groups represented at his church. He expects communication difficulties that may slow down the work as many of the more recent immigrants do not yet speak German well. He is willing to work slowly and deliberately to truly bring diverse perspectives into church planning. Also here, an interesting concept is being proposed that may be important for the Syriac Orthodox Church in the future, should they seek to reach out to their German, Turkish, Kurdish, Russian, and Afghan (and from other nationalities) neighbors. Right now, this willingness does not exist in the Syriac Orthodox Church, which in part is due to the strong language and ethnic entrance blocks to outsiders joining the church.

Another Evangelical pastor and a youth leader in another FEG church voiced interest but showed a lack of understanding for the community. However, upon further investigation, the interest seemed to be more verbal agreement with the research interest at hand than a genuine desire to understand or engage the Aramean community. An effort was made at some point years ago, but it did not seem to provide a true benefit. Much of this was the difference in Evangelical vs. Orthodox understanding and apparent cultural differences. Additional conversations are planned with these Evangelical communities and others in Germany to understand whether youth activities might be cross-denominational at some point.[23] A closer look is needed at how the deployment of monitors to display lyric texts in Evangelical churches may be leveraged in Syriac Orthodox churches to help younger members follow the liturgy if they are lacking language skills. A Syriac Orthodox church in Santee, CA, uses this very successfully for their Arabic-speaking parish where their youth is facing very similar issues of losing the language skills. The liturgy in Syriac is transliterated, while two other lines on the screen show the Arabic and English translations.[24] This may be a very workable model to bring technology into German Syriac Orthodox churches for the benefit of both younger and older attendees who lack the language skills.

A conversation with an Aramean man who had become an Evangelical showed the many tensions between the two communities of Evangelical and Syriac Orthodox. He spoke of physical threats to him when teenage children of Aramean families came to visit his church or, worse yet, were baptized by him.[25] In return, conversations with the Syriac Orthodox faithful show that to them, this is a betrayal of their deepest-held beliefs. There seem to be animosities, especially toward Free Evangelical Churches in Germany, often accused of what American Evangelicals might call "sheep stealing". Syriac Orthodox believe Evangelicalism to be heterodox and lacking. One quoted 2 Thessalonians 3:6 when asked about this issue of former Syriac Orthodox Church members converting to Evangelicalism: "Now we command you, brothers, in the name of our Lord Jesus Christ,

that you keep away from any brother who is walking in idleness and not in accord with the tradition that you received from us".[26] It is a betrayal of their faith, for which their ancestors gave their lives.[27]

In addition, the Syriac Orthodox already have an active youth group within the German diocese, the SOKAD Jugend (Syriac-Orthodox Church Youth in Germany), which operates within the boundaries of their denomination (Erziözese der Syrisch-Orthodoxen Kirche in Deutschland n.d.). The sponsor and founder of this group is the archbishop of Germany, representing the largest diocese in Europe. His Eminence Mor Philoxenus Mattias Nayis (Erziözese der Syrisch-Orthodoxen Kirche in Deutschland n.d.), who resides at the monastery in Warburg in northern Germany, is conscious of the need to train the youth. He has voiced some of the efforts as well as some of the hindrances in an interview with the researcher in October 2022. At the monastery, a weekly class is held for anyone interested in learning the Syriac language and the melodies used in the liturgy. Men, women, and school-age children can participate. He admitted that much of the monastery's work is geared toward educating young men to become altar servers, subdeacons, deacons, and, ultimately, priests. Since the Syriac Orthodox Church does not permit women to serve at the altar, this naturally excludes girls. The SOKAD Youth, targeted at older teenagers and young men and women into early adulthood, seeks to address some of this gender disparity within its programs but cannot offer the richness of programs the monastery can offer to males interested in church service. Mor Philoxenus Matthias Nayis shared one fundamental difficulty for the Syriac Orthodox Church: all other Orthodox faiths have a country where the language is spoken; e.g., Greek Orthodox have Greece, where Greek is still spoken. The Syriac Orthodox faith comprises believers from countries as diverse as Turkey, Syria, Iraq, Lebanon, and Israel. The languages represented make it impossible to use a language other than Syriac for the liturgy. However, when churches have large percentages of a particular immigrant group, elements such as Scripture reading or the sermon may be in Turoyo, Arabic, or German. These portions are small compared to the overall liturgy. The services are approximately 2 1/2 h long, with many families dropping in quite some time past the start time. This is accepted as the morning prayers lead into the liturgy. Children are in the church from a young age and take communion as soon as they are baptized and sealed with Myron oil as infants.[28]

During a conversation with some of the attendees of a luncheon given in honor of a member who had passed away a year prior, a woman shared that she did not understand the liturgy. One young woman, whose family was originally from Syria and spoke Arabic at home, told me that the attendance at this Turoyo-speaking church community was simply a "family outing" to her, as she could not understand. She loves her Aramean community and is very proud of it. Her German is fluent, and she completed studies to be a teacher at a German university, but she cannot follow the liturgy.[29] While preserving the Syriac language in its liturgy is undoubtedly critical, creative ways may be found to help those attending understand their faith tenets. Outside of this, most only learn while in school in Syriac Orthodox *madrasse* or at home if their parents are devout and exercise spiritual disciplines there.

From the Aramean side, there is a lot of interest in having Germans gain a better understanding of their community. The men and women of the Syriac Orthodox Church interviewed were disappointed when German neighbors thought they were Muslims or did not accept invitations to their feast days and ensuing celebrations at the church. Enthusiastic and lengthy replies strongly encourage the liberal sharing of learning about the Aramean community in Germany. When asked what audiences should know about the Aramean people, one woman said, "Tell them the Aramean people live". Conversations and interviews show a lot of pride and hope in the Aramean community. They view their early Christendom faith traditions as beautiful and highlight to the listener that their devotion to God runs deep in their prayer and worship life.[30]

There is, however, a reason for optimism that mutual understanding can be achieved, which will ultimately aid in educating young Arameans in Germany about their faith:

the Lausanne-Orthodox Initiative was established after the Third Lausanne Congress in 2010. It serves as a community of Orthodox and Evangelical Christians. They "wish to respect each other's beliefs, learn from each other, and support one another as we each obey the call to share in God's mission" (Lausanne-Orthodox Initiative n.d.). It will require a getting-to-know each other's belief systems as a first step and eradicating misconceptions on both sides. Participation in the Lausanne-Orthodox Initiative's work can aid in formulating a plan to move forward and bring about an appreciation of the different theological understandings and religious practices. This, in turn, will forge a path toward a collective future while retaining tradition and language in Germany with German neighbors for the now no longer wandering Aramean—and their offspring.

## 13. Conclusions

While much information remains to be gathered for this study, first findings allow some initial assessment of how living in a secular and pluralistic German society has affected the religious development of second and third generations of Aramean families. While children of church-attending families retain close ties to their church community, hindrances are starting to appear in language retention as life for these children centers in the German school systems most days. As more ethnic barriers are broken down through school friendships, German will become the premier language choice for the Aramean youth. Communication between the different parts of church communities will continue to be problematic as Syriac refugees who are Syriac Orthodox are Arabic speakers and do not understand Turoyo for the most part.

The Syriac Orthodox Church in Germany at this point in time is a very homogeneous entity, i.e., the members are of the same ethnic and cultural background, despite some of the recent refugees arriving from primarily Syria. The ancestors of the refugees from Syria were forced to leave the Tur Abdin region and chose at that time to settle their families in Syria. Evangelical communities in Germany have begun to address diversity issues in practical manners to account for the diversity encountered in their churches. Because of the work done in these churches, a number of the issues encountered with the Syriac Orthodox youth could be addressed if a similar approach is applied before these topics potentially turn into insurmountable problems in the future. Having events that target both Syriac Orthodox youth and their friends of other cultural or religious backgrounds may provide a more stable level of interest in the longer term.

At the same time, the Evangelical church in Germany can learn much from the way the Syriac Orthodox parishes build community. Shared engagements, such as celebrations of certain holidays with the church body, would build a stronger cohesion in these Evangelical communities. The encouragement to spiritual disciplines found in the Syriac Orthodox community is another take-away that Evangelical churches might benefit from within their own church settings.

Ultimately, any benefit to be derived will only materialize through consistent and intentional engagement between the communities. This is the catalyst that is still missing.

**Funding:** This research received no external funding.

**Institutional Review Board Statement:** The study was conducted in accordance with the Declaration of Helsinki, and approved by the Institutional Review Board (or Ethics Committee) of Columbia International University (protocol code 238 and 24 March 2022) for studies involving humans.

**Informed Consent Statement:** Informed consent was obtained from all subjects involved in the study.

**Data Availability Statement:** Not applicable.

**Conflicts of Interest:** The author declares no conflict of interest.

## Notes

[1]    Deut. 26:5 (NASB).

2　　Gen. 3 (NASB 1995).

3　　Acts 18:2 (NASB 1995).

4　　An interesting study on religiosity of young refugees by the University of Erlangen-Nürnberg under the leadership of Professor Manfred L. Pirner. A brief report on the pre-study of this research project can be found here: Pirner (2021).

5　　FRG = Federal Republic of Germany.

6　　The information in the previous two paragraphs was captured from a Sayfo recognition flier produced by the Aramean community in Germany for the 100th anniversary of the 1915 massacres in 2015.

7　　From a conversation with an Aramean man in Germany, May 2022.

8　　From a phone conversation with an Aramean man living in Germany since 1980.

9　　Ibid.

10　Some of this can be seen in this blog post from Syriac Press: Beth Shao (2022).

11　See, e.g., Richard G. Kyle, "Nestorius: The Partial Rehabilitation of a Heretic" and Mark Dickens, "Nestorius Did Not Intend to Argue That Christ Had a Dual Nature, but That View Became Labeled Nestorianism".

12　Translated by the author, who is bi-lingual.

13　The schism did not happen overnight. It happened over a period of about 80 years but culminated with the Council of Chalcedon in AD 451.

14　The survey based on established questions was conducted on 12–13 May 2022 as a written, anonymous homework assignment in two German Syriac Orthodox religion classes.

15　From a phone conversation with the head of the Suryoye Kultur- und Sportverein Augsburg e.V., 28 March 2022.

16　From a series of interviews conducted in May and September 2022 in Germany.

17　From the survey of 13–15 year olds already mentioned.

18　Information via input from interviewees in Germany and personal observation.

19　From the survey of 13–15 year olds.

20　Translated by the author.

21　Information from church website at https://www.feg-giessen.de/gottesdienst/simultanuebersetzung/. Accessed on 26 March 2022.

22　Interview with Pastor Torsten Pfrommer on 30 Septermber 2022.

23　From a conversation with Pastor Hauge Burgardt and Kerstin Thielmann at Freie Evangelische Gemeinde in Pohlheim-Watzenborn-Steinberg, Germany, on 15 May 2022.

24　Observed at St. Paul the Apostle Syriac Orthodox Church in Santee, CA, on 20 November 2022 by the author during a visit to the parish.

25　Interview with Aramean Evangelical believer in May 2022.

26　2 These. 3:6 (ESV).

27　Gathered from interviews with Syriac Orthodox lower-level clergy and church members in May 2022.

28　Interview with Mor Philoxenus Mattias Nayis on 3 October 2022.

29　Group observation during a church event in Hanau, Germany, 2 October 2022.

30　From interviews with the Aramean community in May and September 2022.

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
