# Peer review of "A Home for the ‘Wandering Aramean’—In Germany?"

_religions, doi:10.3390/rel13121176_

Round 1

Reviewer 1 Report

This article is interesting mainly because of the information about an ethnic and religious group called Arameans of the Syriac Orthodox Chistianity, who emigrated from south-eastern Turkey, mainly due to persecution, and settled in many European countries, including Germany. The author discusses in detail the historical background of the Arameans' emigration to western Europe and also the dialects of the Aramaic language they use. It is evident that the author knows the Arameans, as he speaks about them with expertise. Other reflections, however, seem rather superficial and not academic. This text is more popular and journalistic than academic. It is not even clear exactly what the academic problem of this article is. If the author's aim was to show the relationship between the Aramean immigrants and the evangelical communities in Germany, it can be said that the goal has not been achieved. The interviews with two pastors who denied such cooperation are not very significant. The question also arises why only the relations of Aramean immigrants with evangelical communities in Germany, mainly the Free Evangelical Church in Gießen, are analysed. What about other faith communities? Also personal references such as 'In my hometown Gießen' or 'when I visit my family' are not used in academic texts. In my opinion, the author should rework the article, but I encourage her/him to resubmit the paper. It can be really interesting. In my opinion it would be better to focus on the religious identity of Aramean expatriates in Germany, identify what problems they face in that secularised country and show some solutions. In this context, it would be worthwhile to do empirical research with representatives of this population, if only a few interviews with Aramean emigrants or their leaders. Personal references should also be avoided.

Author Response

Dear Reviewer, Thank you for the very good pointers you gave me. I have revised the paper significantly based on your and other reviewers' points. The style should be more consistent with what you were expecting.

To answer your question about why only evangelical churches: the Syriac Orthodox Church (based on some of the writing of the German archbishop) has issues with the Evangelical churches in Germany, but not with the mainline state churches. They see many of their young people leaving due to their strong youth programs and evangelistic outreaches. One of the desires behind this research is to give both parties tools to work with each other and benefit.
I would be ecstatic if you liked the changes I made. It is my very first journal submission (I am a PhD candidate).
Thank you so much!

Reviewer 2 Report

Thank you for this paper, which presents the case of Aramean/Suryoye migrants to Germany and their religion. I want to draw your attention to some general issues:

It's basic premise appears to be unfit for an academic paper. You state that "The question to be further explored—possibly within [the Lausanne initiative] context—is how to interest both the Syriac Orthodox believers and the Evangelical community in Germany in a fellow Christian-to-Christian engagement to win their children and grandchildren to the Christian faith." This is question of proselytisation (maybe of interest for some members of either the Syriac or the Evangelical church) but not a question of scientific interest.

Your paper provides plenty of historical and theological background on Arameans/Suryoye people and their religiosity. At times this information appears arbitrarily assembled, rather than following a coherent argumentative strategy. Thereby it is unclear how you went about your research: What was the research question? What was your methodology? Who did you talk to? All this remains unclear. In addition, you do not define central terms, and you do not explain the rationale behind them. E.g. in your abstract you ask for "the impacts" on the life of "families". What do you perceive as an "impact"? Why do you ask for impacts on "families"? Family life does hardly surface in your paper. It remains unclear with whom you spoke in the church and about what you spoke with them.

Surprisingly late in the paper you draw attention to the problem of how people call themselves. You highlight that people do not necessarily perceive themselves as Arameans, that some call themselves Assyrians, Syriacs (which you later use synonymously with Aramean) or Suryoye. You do not provide any explanation for either your preference for Arameans, nor the reasons why (in the 1970s) people started to identify not as one group, but started to make distinctions. Adding to this confusion is that you state in lines 343-344 that both Aramean and Assyrian refer to the same group, while the quote you provide states, that those terms designate two different peoples.

It remains unclear why you write that you are interrested in the impact on Aramean families, yet you hardly write about families, family live, how religion is passed down, the religiosity of the second/third generation etc.

Side note: It is highly likely that over time people will marry outside of their religious community. What happens then? Those would be very interresting questions, which I would expect from this paper given the abstract. However, you do not address them.

Some statements are plainly wrong e.g. that guest workers were needed to build "living spaces quickly after so much destruction" (meaning destruction of WWII). Guest workers were needed due to a labour shortage, since the economy was booming ("Wirtschaftswunder"), not because of rebuilding after WWII (which ended 16 years before the first "Anwerbeabkommen"). The statement that "many Germans are now neighbours with Arameans" is overexaggerating the presence of Arameans, and utterly ignoring processes of inter-religious marriages and other phenomena such as hybrid identities.

At times, you seem to loose the focus of the question you presumably want to discuss, which is the question of "impacts on Aramean people in the diaspora". E.g. it is unclear to me, why you need all this information on Michael the Syrian, and his position towards national identity? It appears to me, that the Michael anecdote does little more than make for an interresting parable. If this is what you intended by it, it is much too long, as it does not add any new information which would help to better understand dominant notions of nation and nationalism among Arameans/Suryoye people in Germany today. (Side note: I would strongly call into question that Michael's concept of nation can be compared to modern notions of nation/national identity.)

In sum, I would strongly advise to rework the paper and decide whether you want to focus on your assessment of family life among Arameans living in Germany today; or the (migration) history of Arameans. In any case state your methodology and formulate a clear research question. Scrutinize statements by people you talk to and don't take them at face value.

For more specific details, please consult the attached PDF.

Reviewer 3 Report

This is a good and very interesting argument that makes a contribution to migration studies as well as religious history.

I think the citations could be improved.  I would suggest at least some citations be added between lines 16 and 65.  While this seems like common knowledge, it is quite specialized compared with what most people know.  

Likewise, there need to be citations lines 79-130, 142-183, 183-191, 220, 282-300, 309-315, 343-348, 361, 369-375, 467-471, 488-534.  In other words, citations need to be regular and consistent -- at least one per paragraph -- to fully reference everything said within the article.  This is my biggest suggestion. 

The article structure and argument are sound.  The subject is innovative, as is the author's approach to it.

As noted, referenced should be added throughout the article.  They may be additional references to works already cited, but the source for all information discussed needs to be clear, with at least one reference per paragraph.

Author Response

Dear Researcher,

Thank you for both your encouraging remarks and your valuable input!
I have added quite a bit of additional information and references since I got to go back for more research since I submitted the proposal.

I trust this has made the article stronger!

Much appreciated!

Round 2

Reviewer 1 Report

Dear author, thank you very much for sending the revised manuscript. You have done good job and added a lot of important infromations. However, in the abstract you write, your paper „will review the impacts on the life of devout Syriac Orthodox families while living in Germany, a secular country. It will also take an initial look at whether evangelical communities in Germany can come alongside this group”. But you devote little space to this problem, in fact only two short paragraphs. Your presentation of this ethical group and language etc. is very interesting, but it would be worthwhile to devote more space to solving the main problem of the paper. Furthermore, the interviews with pastors are very few and do not allow to draw wider conclusions. It would be good if you added a "Conclusion" section at the end of the text where you can synthesise the results of your research.
Thank you again for your contribution.

Author Response

Dear Reviewer,

Thank you for your very well-received comments. 

I have worked on the sections you mentioned. Please note that my dissertation work is still ongoing, which makes some conclusions tentative. Especially interviewing Evangelical pastors has been harder than expected. It took me multiple rejections for an interview and being persistent to get the ones I have. I am not exactly sure why. The Syriac Orthodox clergy has been much more forthcoming. Nevertheless, I feel that I have improved this section based on the most recent input received.

I have also added a number of stronger references as well as a Conclusion section. That was indeed missing - thank you!

I hope this serves to make this article valuable for readers!

Thanks again - your input has given me much to improve on!

Reviewer 2 Report

The article has improved significantly since the initial review. The structure is much easier to follow, many ambiguities (e.g. regarding the methods) have been cleared up and questions have been answered.

Still, I would not refer to it as a work applying a grounded theory approach; for this there is not nearly enough empirical data presented or discussed;in fact most of the arguments rely on an analysis of secondary material. With this in mind, I suggest to revise the second paragraph:

x) "diaspora experience" is not the focal point of the paper, but provides an overview of the origin and history of the Aramean people and the Syriac-Orthodox church, as well as the migration patterns and religio-cultural adaption processes in the context of the Aramean/Syriac diaspora in Germany. (Those suggestions may be used/adapted by the author.)

x) this is not a paper applying grounded theory but rather it works with a general hermeneutic approach, which is applied to different historical and secondary sources. The aim seems to be to provide a comprehensive narrative on the history of an ethno-religious group (Arameans) over time with special attention to the migraton process and diaspora situation in Germany. (Those suggestions may be used/adapted by the author.)

x) It appears to me that the (expert?) interviews have only provided additional information. This should be clearly stated.

Moving on, I would suggest to review the text and get rid of repetitions (e.g. on lines 201-203). In general, the article could be a bit more concise.  Furthermore references should be checked if they are following the style guide of the journal (many years of publication, etc. seem to be missing).

The title of the chapter on national identity refers to "early and modern thought" - I am no historian, but "early thought" appears an odd choice of words to me. Check if that is the apropriate term for what you mean.

It appears to me that you want to make the case that national identity is closely tied to religious identity; this point I would emphasise more strongly at the beginning of the chapter on national identity. In literature on nationalism, it is common to think of a common language as the origin and most important pillar of national identity. Here you tell a different story. You should consider referring to Benedict Anderson's work Imagined Communities at this point, as he explicitly discusses the question of national identity and its relation to religion and language.

Author Response

Dear Reviewer,

I am so amazed at the truly excellent feedback you provide. Very helpful to sharpen the content.

Diaspora experience is certainly a very critical element of identifying national identity for the Aramean people I encounter, but right now, you are absolutely right that I describe how they got there and early adaptation to their new environment. I have deleted the term and rather inserted your comments.

While ultimately my dissertation will be based on grounded theory, I agree that this is misleading in the current status of my research. As such, I agree that "selling it" a grounded theory for this article is wrong. I have taken most of your wording to heart and added it.

I have softened the wording on the interviews I conducted. I have done many more than the anecdotes suggest. However, most of these need to still be transcribed and analyzed.

I got rid of the repetition you noted. I am struggling to make the article more concise as the elements looked at all bring to bear where these dear people find themselves in 2022. 

I checked the style guide, and "n.d." seems to be the recommended term if no date is provided on a website while offering the access date in the bibliography. I have also some references with stronger examples to avoid that problem! Nevertheless, some of the data is only found on the diocese websites or public pages (tax status, etc.). None of these have a date when the content was created. This is what makes websites tricky...!

I fixed the title of national identity chapter. That was the German coming through! 

I have ordered Anderson's book, however, I do not have it yet. I had so far focused on Anthony D. Smith's works. I have used a point he makes to get to the language element. I am grateful for the recommendation of Benedict Anderson's work. 

I also added a Conclusion section to summarize. One reviewer felt this was missing.

I am extremely grateful for all your input. I wish I could keep sending you pieces of my dissertation! You are by far the best reviewer out of the three I had. I wish I could keep you!

Kind regards,